# Percutaneous Extraction of Transvenous Permanent Pacemaker/Defibrillator Leads—A Single-Center Experience

**DOI:** 10.3390/medicina60081360

**Published:** 2024-08-21

**Authors:** Murat Akcay, Serkan Yuksel

**Affiliations:** Department of Cardiology, Ondokuz Mayıs University School of Medicine, Samsun 55139, Turkey; serkany77@yahoo.com

**Keywords:** transvenous lead extraction, indications, outcomes, single-center experience

## Abstract

*Background and Objectives*: The number of cardiac pacemakers being used has increased in recent decades, and this increase has led to a rise in device-related complications, requiring percutaneous device extraction. Our aim was to present our single-center clinical experience in percutaneous lead extractions. *Materials and Methods*: We analyzed an observational retrospective cohort study of 93 patients for the transvenous removal of a total of 163 endocardial leads. We evaluated the device details, indications, lead characteristics, extraction methods, complications, reimplantation procedure, follow-up data, effectiveness, and safety. *Results*: Patients’ mean age was 68.6 ± 11.6 years. Lead extraction indications were pocket infection in 33 (35.5%), lead dysfunction in 33 (35.5%), and system upgrade in 21 (23%) cases, and lead endocarditis in 6 (6%) cases. The duration from implantation to extraction time was a detected median of 43 (24–87) months. The most common retracted lead type was the RV defibrillator lead (62%), and the lead fixation type was active for one hundred (61%) patients. A new device was inserted in 74 (80%) patients, and the device type was most commonly a CRT-D (61%). Patients were followed up at a median of 17 (8–36) months, and 18 patients (19%) died at follow-up. Complete procedural success was obtained in 78 (84%) patients, and clinical procedural success was obtained in 83 (89%) patients. Procedural failure was detected in 15 (16%) patients. Major and minor complications were detected in 10 (11%) and 6 (6.5%) patients, respectively. The most common minor complication was pocket hematoma. *Conclusions*: Our experience suggests that transvenous lead extraction has a high success rate with an acceptable risk of procedural complications. The simple manual traction method has a high rate of procedural success, despite a high dwell time of the lead.

## 1. Introduction

The number of cardiac pacemakers being used has increased in recent decades, according to the suggestions in guidelines and the proven influence of cardiac resynchronization therapy. This increase has led to a rise in device-related complications, such as infection and lead dysfunction, requiring device extraction [1,2,3]. The number of pacemaker- and lead-related problems requiring lead extraction, including infection, lead dysfunction, venous stenosis, thrombosis, lead-associated arrhythmias, and the need to upgrade to a new device system, is also increasing [1,2,3]. After pacemaker implantation, leads mostly undergo fibrotic encapsulation in the venous system by the activation of various cellular and humoral immune processes. Although the manual extraction method may be a potent technique to remove leads implanted <1 year ago, more advanced extraction systems are required for chronically implanted leads due to fibrous adhesions and encapsulations [3]. Mechanical extraction systems have been developed rapidly, but there are great differences in the experiences of different clinics due to the accessibility and high costs involved in using these systems. Despite all the technical advances, lead extractions remain a compelling, dangerous procedure and require surgical back up [4]. However, recent studies have shown that discharge on the same day is safe after uncomplicated lead extraction, completed in the morning in patients without device infection [5].

In this study, we present our clinical experience, in terms of lead dysfunction type, clinical profiles, lead extraction indications, methods used, complications, follow-up data, effectiveness, and safety.

## 2. Materials and Methods

### 2.1. Study Protocol

We present an observational retrospective analysis of 93 patients, aged between 42 and 94 years, who underwent transvenous lead extraction procedures in a tertiary center from January 2014 to December 2019 for the transvenous extraction of a total of 163 endocardial leads. Patient data, device details, indications, lead types, lead characteristics, complications, reimplantation procedure, and follow-up data were provided from the electronic archive data of the patients. For all patients, age, sex, cardiovascular risk factors, and NYHA functional status were recorded in patient forms. Procedural details were also collected from patient reports. This study was approved by the Ethics Committee of the Faculty of Medicine, Ondokuz Mayis University (No: 2021/385) and adhered to the Declaration of Helsinki (2013 version).

### 2.2. Lead Extraction Technique

All of the procedures were conducted by two operators (S.Y. and M.A.). The lead removal indications were based on the guidelines, and the main indications were lead dysfunction, device infection (pacemaker pocket infection or lead endocarditis), and pacemaker upgrade [4,6]. The lead extraction operation was performed in the cardiac electrophysiology laboratory under conscious sedation and local anesthesia, monitoring non-invasive blood pressure, electrocardiography, and pulse oximetry. A complete analysis of devices was performed before the extraction procedures, including evaluating patient needs for a pacemaker, and if necessary, a temporary transvenous pacemaker was inserted. We usually applied pre-procedural venography before lead extraction to determine the appropriate technique to use and to reduce venous complication rates. After skin preparation, the device pocket was opened without damaging the leads, and the leads were disconnected from the generator. If appropriate, primarily dysfunctional leads were split from the scar tissue without damaging other leads by blunt dissection. Then, the leads were extracted using the simple manual traction method, rotating mechanical dilator sheath, and snare loop catheter techniques.

### 2.3. Definitions

Definitions were used based on the 2018 EHRA expert consensus declaration on lead extraction document (2018) [6]. The simple manual traction method for lead extraction is often preferred as the primary approach for removing the pacemaker lead, as in our clinic. The simple manual traction technique was used as the first method, especially in active leads and short dwelling-time leads. The rotating mechanical dilator sheath (Evolution^®^, Cook Medical, or TightRail™, Spectranetics, Colorado Springs, CO, USA) with a locking stylet was used as a mechanical extraction tool. A transfemoral approach was only used to remove the leads or lead fragments when subclavian attempt techniques failed. A snare loop catheter was always used to catch the leads or lead fragments. If snare loop catheters failed, a 4 mm steerable ablation catheter was inserted to free the lead ends through the use of a snare loop catheter [6,7,8,9]. Laser and electrosurgical sheaths were not used. We could not use special locking-style guides and laser-energized release sheaths due to difficulties in obtaining materials.

### 2.4. Definition of Success

#### 2.4.1. Complete Operative Success

Removal of all targeted leads and materials without permanently disabling complications or procedural death.

#### 2.4.2. Clinical Procedural Success

A small portion of a lead remaining that does not adversely affect the outcome goals of the procedure. This may be the terminal or small portion (<4 cm) of the lead (conductor coil, insulation, or both) if the remaining portion does not increase the risk of complications such as perforation, embolization, and infection, or cause any adverse outcome. There are no disabling complications or operation-associated death.

#### 2.4.3. Operative Failure

Insufficiency to reach either exact operative or clinical success, or the improvement in any persistently disabling complication or operation-associated death.

#### 2.4.4. Complete Lead Extraction

Complete removal of all targeted lead and materials.

#### 2.4.5. Incomplete Lead Extraction

Partial lead extraction where part of the lead remains in the patient’s body (vascular or extravascular).

### 2.5. Description of Complications

#### 2.5.1. Intraoperative Complications

Any condition related to the events of an operation that obviously occurs from the period when the patient enters the operating area until the patient exits from the operating area. This includes complications associated with the preparation of the patient, providing anesthesia, and from the beginning to the end of the operation.

#### 2.5.2. Early Postoperative Complications

Any state associated with the operation that happens to manifest itself within 30 days following the intraoperative period.

#### 2.5.3. Late Postoperative Complications

Any state associated with the operation that happens to manifest after 30 days following the intraoperative duration and during the first year.

#### 2.5.4. Major Complications/Critical Adverse Results

Any of the results associated with the operation, which is life-threatening or results in death (cardiac or non-cardiac). Moreover, any unforeseen state that causes permanent or significant disability needing in-patient hospitalization or an extension of the stay in hospital, or any condition requiring a major surgical operation to avoid any of the results recorded here.

#### 2.5.5. Minor Complications

Any unwanted status associated with the operation that needs medical intervention or a minor operation for therapy and that does not permanently or significantly limit the patient’s life, threaten life, or cause death [6,7,8,9,10].

### 2.6. Statistical Analysis

All research parameters were recorded in the computer and assessed by “The JAMOVI project (2020), JAMOVI (Version 1.1.9.0)”. The suitability of the parameters to normal distribution was evaluated by visual (histogram and probability graphs) and analytical techniques (Kolmogorov–Smirnov test). Continuous parameters were submitted as mean ± standard deviation (SD) if the data were normally distributed, otherwise median (interquartile range as 25th and 75th percentiles) values were given. Categorical variables were represented as numbers and percentages. *p*-values less than 0.05 were accepted as statistically important.

## 3. Results

From January 2014 to December 2019, a total of 163 endovascular leads were removed from 93 patients who were admitted to our cardiology unit. Sixty-seven patients (72%) were male, and the mean age was 68.6 ± 11.6 years. Twenty-eight patients had atrial fibrillation (30%), seventy-four patients had heart failure (80%), and the mean ejection fraction was 35.3 ± 13.2 (%). Twenty-two patients (24%) had a history of previous battery replacement. The median interval between battery replacement to lead extraction time was detected as 30.5 (8.75–69.5) months. Baseline descriptive characteristics are shown in Table 1.

The device types and indications for implantation of the first device in patients are presented in Table 2. It was observed that the first device type was most frequently implanted for CRT-D (31 patients, 33%) and cardiac resynchronization treatment indication (33 patients, 35%). The indications for lead extraction were device pocket infection in 33 (35.5%) cases (Figure 1), lead dysfunction in 33 (35.5%) cases, device upgrade in 21 (23%) cases, and lead endocarditis in 6 (6%) cases. The median time from implantation to extraction was detected as 43 (24–87) months or 3 (2–7) years. The retracted number of leads per patient was detected as a median of 2 (1–2) and a total of 163 leads. The most commonly retracted lead type was the RV defibrillator lead (62%), and one hundred (61%) lead fixation types were active. The mechanical extraction system was used in only five patients (5%) due to difficulties in availability and high cost. The techniques and lead characteristics in the patients are shown in Table 3 and Figure 2.

After lead extraction, the new device was inserted in seventy-four (80%) patients. After lead extraction, the new device was implanted at the same session in forty-nine patients (66%). The pacemaker was implanted in the same pectoral region in thirty patients (61%) and in the opposite pectoral region in nineteen patients (39%). In all patients, the new device implantation was performed in the same or opposite pectoral region according to the clinical features or extraction indications of the patients. The median reimplantation time was detected as 3 (0–8) days and the most common implanted device type was a CRT-D (61%). After lead extraction, the patients were followed up for a median of 17 (8–36) months or 1 (0–2.75) year and eighteen patients (19%) died during follow-up. The total duration of follow-up after implantation for 91 patients was a median of 66 (46–108) months or 5 (3–9) years. The median duration of hospitalization for extraction was detected as 6 (3–14) days (Table 4). Complete operative success was achieved in 78 (84%) patients and clinical operative success was achieved in 83 (89%) patients. Procedural failure was detected in 15 (16%) patients. The major and minor complications were detected in 10 (11%) and 6 (6.5%) patients, respectively. Major complications were intraoperative right ventricular rupture and death in one patient, postprocedural in-hospital cerebrovascular stroke in two patients, and bleeding requiring three or four units of blood replacement and pocket hematoma in five patients. Three patients died during in-hospital follow-up with symptoms and signs of advanced heart failure without pacemaker complications. All minor complications were pocket hematoma without significant bleeding (in six patients). The details are shown in Table 4.

## 4. Discussion

We present our single-center experience with lead extraction. The indications for extraction in our center were similar to other registry studies or center experiences [2,3,6,7,8,9]. The median duration or age of the leads was 3 (2–7) years, partially longer than in other studies [2,3,6,7,8,9,10]. Again, the basic manual traction method was more common (85%) as an extraction technique. Reimplantation of a new device after lead extraction was higher (80%) and the CRT-D was also high (61%). Finally, the clinical procedural success rate in our center was 89%.

In recent years, the frequency of lead removal and clinical experiences has raised as a result of the increasing prevalence of patients with pacemakers. Reported outcomes show high success rates and serious operative complications are limited. Vascular avulsion or tear, superior vena cava laceration and cardiac rupture, tamponade are the major mortal complications of lead extractions [6,7,8,9]. The minor complication ratios of transvenous lead extraction are reported differently in the literature. In our study, the rate of pocket hematoma (minor complication) was 6.5%. Zabek et al. [1] reported a minor complication rate of 3% and Bongiorni, et al. [11] reported 7.2%. The major complication ratio was higher (11%) in our study, when compared to other studies [6,7,8]. In the ELECTRA (European Lead Extraction ConTRolled Registry) study displayed that obstruction or severe stenosis of the superior venous entry was shown to be an independent predictor of cardiac complications [7]. According to published previous data’s and the ELECTRa registry showed that the patient factors (such as comorbidities), device/lead features, lead fixation types such as active-passive fixation as well as the center-related factors (high or low volume centers, operator experiences) were related with both lead extraction success rate and major complications [8,12,13,14,15,16,17,18,19,20,21].

The first large prospective controlled registry, the ELECTRa registry showed the most common indication for lead extraction was infection (52.8% of all cases) and the second most common indication was lead malfunction [8]. Our results were consistent with this study. However, after infection and lead dysfunction, system upgrade (23%) was found to be high in our study. This can be explained by the development of heart failure over time and the change in indications in the guidelines. The most common device type reimplanted was CRT-D (61%), in our study.

The large variability in the success rate of the basic traction method was detected in among the different studies. In an Indian study, lead extraction of predominantly pacemaker leads, using simple manual traction methods were successful in removing three-fourths of electrodes [22]. In a Brazilian study with 35 patients, the median age of the leads was 46.22 months, simple traction was used in 88.9% of the procedures and complete extraction was achieved in 90% of the cases [23]. The lead characteristics, differences in lead age, extraction indications and center experiences are the main factors affecting the success rate [13,17,18,19,20,21]. In our study, although the lead age was high (median 3 (2–7) years), the extraction rates were high (in seventy-nine patients, 85%) with the basic manual traction method. In the ELECTRa study, 3510 patients were included, and complete lead extraction was obtained in 1743 (93.5%) patients, partial (<4 cm left) in 88 (4.7%) patients, and unsuccessful (>4 cm left) lead extraction in 32 (1.8%) patients [10]. In our study, complete operative success was achieved in 78 (84%) patients, clinical operative success in 83 (89%) patients and procedural failure in 16 (17%) patients.

The risks of transvenous lead extraction are varied in ICD leads compared to single or dual coils. The superior vena cava (SVC) coil is frequently located in a high-risk area, enable fibrous tissue ingrowth and related with significantly higher complication rates. In the literature, the use of powered sheath was more widespread in dual coil leads compared with single-coil ICD leads [17,18]. In our study, dual coil ICD lead extractions (33%) were the most commonly extracted lead type, but our powered sheath usage rate was low due to high cost.

The mean age of all 163 extracted leads (62.7 ± 58.2 months) was partially high in compared to literature data. Zabek, et al. [1] reported a mean age of 76.2 ± 60.2 months, Bongiorni et al. [11] reported a mean age of 69.3 months, Kennergreen et al. [14] reported a mean age 69.0 months and Kutarski et al. [15] reported a mean age of 76.7 months. Again, Pecha et al. [24] showed that lead extraction in leads over 10 years of age is safe and effective using of different extraction tools in specialized centers. It is needed to emphasize that despite the high age of the leads and the high rate of use of manual traction method, we reached a high rate of operative success (89%) comparable with the outcomes of other studies [12,13,14,15,16,17,18,19,20,21,22,23].

After lead extraction, the rate of new device implantation has been changed in the literature [25,26]. Barakat et al. [27] showed that despite the high prevalence of heart failure and comorbidities, lead extractions for device upgrade are related with high success rates and low major complication rates, especially in high-volume centers. Another study showed that fourteen percent of patients were not implanted the new device after extraction due to change in indication or other reasons [26]. In patients who are not reimplanted, the average life expectancy is significantly lower with respect to device-related complications and comorbid conditions, but arrhythmia-related death is rare. Consequently, all factors should be evaluated in detail, including the cause for extraction, the indication for new device implantation, the patient wishes, the reimplantation device type, location, and timing and the requirement for a bridging device [4,10,25,27]. In our study, after lead extraction, the new device was implanted in seventy-four patients (80%) and in forty-nine patients (66%) at the same session ratio.

Although lead extraction operative mortality is extremely low, especially in high-capacity clinics, post-procedural and long-term mortality remain high in specific patients, especially those undergoing lead extraction for infectious indications and device upgrades [8,10,20]. The study of Maytin et al. [28] showed that in 985 patients who underwent 1043 lead extraction procedures, the mortality rate was 2.1% at 30 days, 4.2% at 3 months, and 8.4% at 1 year. The study of Grammes et al. [29] showed a mortality rate of 10% at 30 days. The study of Henrikson et al. [30] showed a mortality ratio of 30% during the follow-up time (range, 6–55 months). There were no operation-associated deaths, and the all-cause mortality risk was 44% for systemic infection. In our study, we detected the death ratio during follow-up was 19%. Our follow-up rates were 1 (0–2.75) year after extraction and 5 (3–9) years after implantation total duration.

### Study Limitations

Our study is a retrospective observational study, subject to bias and the other restrictions of non-experimental studies. As a single tertiary reference center study with a relatively small sample volume, its generalizability is limited. Nevertheless, it is very informative in terms of clinical profiles, techniques used, lead characteristics, indications, success rates, and long follow-up times and the results are similar to other studies. Patient mortality rates were obtained from the national death registration system. Therefore, the inability to know the causes of death of our patients during the long follow-up period is another limitation of our study.

## 5. Conclusions

Our experience suggests that transvenous lead extraction has a high success rate with an acceptable risk of procedural complications. The simple manual traction method has a high procedural success rate, despite the old age of the lead. According to our data and the available literature, we can conclude that transvenous lead extraction, when performed by experienced operators, using normal procedures, is a safe and feasible therapy with few procedure-related complications and rarely life-threatening complications.

### Highlights

(1)The number of cardiovascular implantable devices has increased significantly over the last decades and this increase has led to an increase in device-related complications, such as infection and lead dysfunction, requiring device extraction.(2)We presented our single-center clinic experience with an observational retrospective cohort study of 93 patients for the transvenous extraction of a total of 163 endocardial leads.(3)Our experience suggests that transvenous lead extraction has a high success rate with an acceptable risk of procedural complications.(4)The simple manual traction method has a high rate of procedural success rate, despite the old age of the lead.(5)Transvenous lead extraction, when performed by experienced operators, using normal procedures, is a safe and feasible therapy with few operative complications and rarely life-threatening complications.

## Figures and Tables

**Figure 1 medicina-60-01360-f001:**
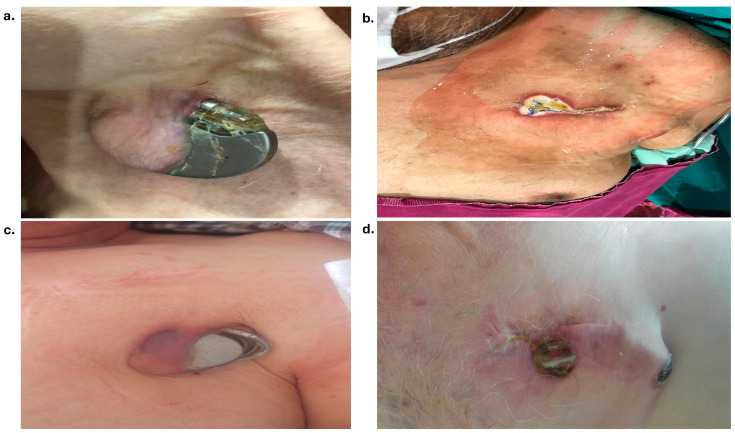
(**a**–**d**) Imaging of pacemaker pocket infection or erosion of different patients who underwent lead extraction due to pacemaker pocket infection.

**Figure 2 medicina-60-01360-f002:**
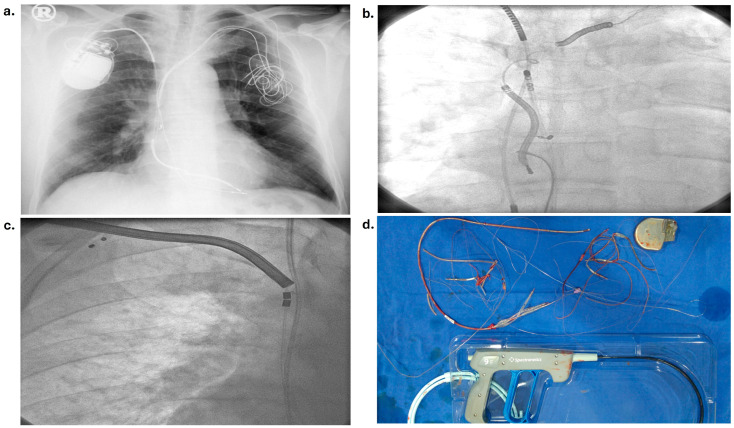
(**a**) Imaging of chest radiography of a patient who will undergo lead extraction; (**b**) angiographic imaging of a patient who underwent lead extraction using a snare loop and ablation catheter; (**c**,**d**) imaging of the patient who underwent difficult lead extractions using rotating mechanical dilator sheath and mechanical extraction tool.

**Table 1 medicina-60-01360-t001:** Baseline characteristics of the study population.

Parameters	*n* = 93 (%) Mean ± SD; Median (IQR)
Age, years	68.6 ± 11.6
Gender—Men *n* (%)	67 (72%)
Women *n* (%)	26 (28%)
Coronary artery disease, *n* (%)	43 (46%)
Diabetes mellitus, *n* (%)	33 (35%)
Congestive heart failure, *n* (%)	74 (80%)
Atrial fibrillation, *n* (%)	28 (30%)
LVEF (%)	35.3 ± 13.2
NYHA Class I–IV	
Class I–II, *n* (%)	48 (52%)
Class III, *n* (%)	45 (48%)
Previous battery replacement, *n* (%)	22 (24%)
Interval between the battery replacement and lead extraction time (month)	30.5 (8.75–69.5)

**Table 2 medicina-60-01360-t002:** Types of device and indications for implantation of first device in patients.

Type of First Device Implanted	*n* = 93 (%)
CRT-D	31 (33%)
ICD-VR	24 (26%)
DDD	18 (19%)
ICD-DR	8 (9%)
VVI	7 (8%)
VDD	3 (3%)
CRT-P	2 (2%)
**Indications for first device implantation**	***n* = 93 (%)**
Cardiac resynchronization therapy	33 (35%)
Primary prevention, ICD	23 (25%)
Third degree AV block	19 (20%)
Secondary prevention, ICD	9 (10%)
Sick sinus syndrome	9 (10%)

Cardiac resynchronization therapy-defibrillator (CRT-D); cardiac resynchronization therapy-pacemaker (CRT-P); dual-chamber, pacing, and sensing the atria and ventricle pacemaker, two leads (DDD); implantable cardioverter-defibrillator (ICD); -VR (ventricular, single lead), -DR (atrial and ventricular, dual chamber, two leads); isolated ventricular pacing, single chamber, single lead (VVI); single chamber pacing, dual chamber sensing, single lead (VDD).

**Table 3 medicina-60-01360-t003:** Characteristics of patients undergoing lead extraction.

Indications for Lead Extraction	*n* = 93 (%)
Lead dysfunction	33 (35.5%)
Pacemaker pocket infection	33 (35.5%)
System upgrade	21 (23%)
Lead endocarditis	6 (6%)
**Characteristics of patients undergoing lead extraction**	**Median (IQR)**
Month	43 (24–87)
Year	3 (2–7)
**Retracted lead characteristics**	* **n** * ** = 163** **Median (IQR)**
Retracted lead number	2 (1–2)
**Number of leads per patient**	***n* = 93 (%)**
1 lead	46 (49%)
2 leads	25 (27%)
3 leads	21 (23%)
4 leads	1 (1%)
**Lead types**	***n* = 163 (%)**
Atrial lead	45 (28%)
RV pace lead	29 (18%)
RV defibrillator lead	58 (35%)
Single coil	4 (2%)
Double coil	54 (33%)
LV lead	31 (19%)
**Lead fixation**	***n* = 163 (%)**
Active lead	100 (61%)
Passive lead	63 (38%)
**Extraction technique**	***n* = 93 (%)**
Simple manual traction method	79 (85%)
Use of extraction system	5 (5%)
Use of snare loop	9 (10%)

**Table 4 medicina-60-01360-t004:** After lead extraction, reimplantation of new device, follow-up characteristics, complications, success rate, and other features in the patients.

New Device Reimplantation after Lead Extraction	*n* = 74 (80%)
Reimplantation at time of extraction (at the same session)	49 (66%)
same pectoral area	30 (61%)
other pectoral area	19 (39%)
Reimplantation day after extraction, day Median (IQR)	3 (0–8)
**Frequencies of implanted device types after extraction**	***n* = 74 (%)**
CRT-D	45 (61%)
DDD	10 (14%)
ICD-VR	8 (11%)
ICD-DR	4 (5%)
VVI	4 (5%)
CRT-P	2 (3%)
VDD	1 (1%)
**Follow-up after extraction**	* **n** * ** = 93** **Median (IQR)**
Follow-up after extraction, (month)	17 (8–36)
(year)	1 (0–2.75)
Death during follow-up, *n* (%)	18 (19%)
**Total follow-up after implantation**	***n* = 91**
Total duration of follow-up after implantation (month)	66 (46–108)
(year)	5 (3–9)
**Extraction data of patients**	* **n** * ** = 93** **Median (IQR)**
Duration of hospitalization for extraction (day)	6 (3–14)
**Success and complications**	***n* = 93 (%)**
Complete procedural success	78 (84%)
Clinical procedural success	83 (89%)
Procedural failure	15 (16%)
Complete lead removal	85 (%91)
Intra-procedural complications	1 (1%)
Early post-procedure complications	9 (10%)
Late post-procedure complications	7 (7.5%)
Major complications	10 (11%)
Minor complications	6 (6.5%)

Cardiac resynchronization therapy-defibrillator (CRT-D); cardiac resynchronization therapy-pacemaker (CRT-P); dual-chamber, pacing and sensing the atria and ventricles pacemaker, two leads (DDD); implantable cardioverter-defibrillator (ICD); -VR (ventricular, single lead); -DR (atrial and ventricular, dual chamber, two leads); isolated ventricular pacing, single chamber, single lead (VVI); single chamber pacing, dual chamber sensing, single lead (VDD).

## Data Availability

The data presented in the study are included in the article material, further inquiries can be directed to the corresponding author.

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
