# Peer review of "Percutaneous Extraction of Transvenous Permanent Pacemaker/Defibrillator Leads—A Single-Center Experience"

_medicina, 2024, doi:10.3390/medicina60081360_

Round 1

Reviewer 1 Report

Comments and Suggestions for Authors

Authors must be congratulated for having shared their single center experience as transvenous lead extraction is becoming one of the most relevant, and challenging, procedure in cardiology. The ide behind the paper is interesting, but authors need to improve and expand some sections:

Authors should include an image of a TLE, better if one of particular difficulty

Lines 80-90 should be put in lead extraction technique

How many leadless pacemakers did authors implant after TLE?

Specify how many device were implanted in the same side of the extracted ones.

Authors should use different statistical methods to compare the proportions of success rates, complication rates, and re-implantation rates between the basic manual traction method and other extraction methods; they also might perform logistic regression to evaluate the association between the type of extraction method and the likelihood of procedural success or complications, indeed adjusting for potential confounding variables like patient age, gender, comorbidities, and lead characteristics. Yet, authors should do to Kaplan-Meier Survival analysis for  comparing the survival rates between groups of patients who underwent different extraction methods, followed by a log-rank test to assess statistical significance.

Moreover, in order to improve your introduction regarding the feasibility of TLE, I suggest to include and discuss: Safety and feasibility of same-day discharge following uncomplicated transvenous lead extraction. J Cardiovasc Electrophysiol. 2024 Feb;35(2):278-287. doi: 10.1111/jce.16147. Epub 2023 Dec 10. PMID: 38073051.

Comments on the Quality of English Language

minor english revisions are needed

Author Response

Reviewer 1

Comments and Suggestions for Authors

Authors must be congratulated for having shared their single center experience as transvenous lead extraction is becoming one of the most relevant, and challenging, procedure in cardiology. The ide behind the paper is interesting, but authors need to improve and expand some sections:

  1. Authors should include an image of a TLE, better if one of particular difficulty
  2. Lines 80-90 should be put in lead extraction technique
  3. How many leadless pacemakers did authors implant after TLE?
  4. Specify how many devices were implanted in the same side of the extracted ones.5
  5. Authors should use different statistical methods to compare the proportions of success rates, complication rates, and re-implantation rates between the basic manual traction method and other extraction methods; they also might perform logistic regression to evaluate the association between the type of extraction method and the likelihood of procedural success or complications, indeed adjusting for potential confounding variables like patient age, gender, comorbidities, and lead characteristics. Yet, authors should do to Kaplan-Meier Survival analysis for comparing the survival rates between groups of patients who underwent different extraction methods, followed by a log-rank test to assess statistical significance.
  6. Moreover, in order to improve your introduction regarding the feasibility of TLE, I suggest including and discuss: Safety and feasibility of same-day discharge following uncomplicated transvenous lead extraction. J Cardiovasc Electrophysiol. 2024 Feb;35(2):278-287. doi: 10.1111/jce.16147. Epub 2023 Dec 10. PMID: 38073051.
  7. Comments on the Quality of English Language minor english revisions are needed

Response to Reviewer 1

  1. In accordance with your suggestions, we added the two pictures of difficult lead extraction to our article.
  2. You are right. We added an additional section to the lead extraction techniques section. However, we did not change the definitions section because there are specific guideline definitions.
  3. We have never used a leadless pacemaker due to economic opportunities.
  4. Thank you for your additions. We added to the article the number of removed leads inserted into the same and opposite pectoral regions in the same session.
  5. You are right. But our study is mainly a descriptive study and the number of patients is small. We received professional statistical support. We concluded that the number of patients and the distribution of the data were not sufficient for further statistical analysis. The distribution of extraction methods (such as Simple manual traction method 79 (85%), Extraction system use 5 (5%); Snare loop use 9 (10%)) is not sufficient for statistical comparative analysis. The results that can be obtained with advanced statistical analysis by increasing the number of patients in the future will be exciting.
  6. We read the article you suggested and discussed and included it in our article.
  7. We re-read our article, received professional language support and made the necessary corrections in the English language and flow of the article.

Thank you very much for taking the time to review this manuscript. Please find the detailed responses below and the corresponding revisions/corrections with red highlighted/in track changes in the re-submitted files.

We have indicated the corrections in red.

Thank you for your contributions.

Reviewer 2 Report

Comments and Suggestions for Authors

The authors present a single-center observational data of 93 patients that underwent percutaneous endocardial lead extractions.  Patients were followed up for a median of 17(8-36) months and eighteen patients (19%) died at follow-up. Complete procedural success was obtained in 78(84%) patients and clinical procedural success was obtained in 26 83(89%) patients. The authors conclude that transvenous lead extraction has high success rate with an acceptable risk of procedural complications. 

The manuscript is generally well written. The tables are detailed and well-organized.

What I am missing in the manuscript is figures. I believe at least one figure has to be provided for better illustration of the results and methods of the study. For example, a figure illustrating the method of lead extraction or an example case may be a nice addition to the manuscript.

Author Response

Reviewer 2

Comments and Suggestions for Authors

The authors present a single-center observational data of 93 patients that underwent percutaneous endocardial lead extractions.  Patients were followed up for a median of 17(8-36) months and eighteen patients (19%) died at follow-up. Complete procedural success was obtained in 78(84%) patients and clinical procedural success was obtained in 26 83(89%) patients. The authors conclude that transvenous lead extraction has high success rate with an acceptable risk of procedural complications. 

The manuscript is generally well written. The tables are detailed and well-organized.

What I am missing in the manuscript is figures. I believe at least one figure has to be provided for better illustration of the results and methods of the study. For example, a figure illustrating the method of lead extraction, or an example case may be a nice addition to the manuscript.

Response to Reviewer 2

Thank you very much for taking the time to review this manuscript. Please find the detailed responses below and the corresponding revisions/corrections with red highlighted/in track changes in the re-submitted files.

In accordance with your suggestions, we added the two pictures of difficult lead extraction to our article.

We have indicated the corrections in red.

Thank you for your contributions.

Round 2

Reviewer 1 Report

Comments and Suggestions for Authors

Congratulations to the authors for having improved so much their manuscript. I am totally satisfied with this newer version of the paper.